# An Intense Bout of Acute Aerobic Exercise, but Not a Carbohydrate Supplement, Improves Cognitive Task Performance in a Sample of Black, Indigenous, and People of Color (BIPOC) Student Athletes

**DOI:** 10.3390/sports11040088

**Published:** 2023-04-20

**Authors:** Megan Sax van der Weyden, Joel Martin, Jose Rodriguez, Ali Boolani

**Affiliations:** 1Sports Medicine Assessment Research & Testing (SMART) Laboratory, George Mason University, Manassas, VA 20110, USA; 2Department of Public Health, Yale University, New Haven, CT 06511, USA; 3Honors Program, Clarkson University, Potsdam, NY 13699, USA

**Keywords:** cognitive function, maximal aerobic exercise, athletes, performance, carbohydrate rinsing

## Abstract

Background: There are contradictory findings in the literature on whether an acute bout of aerobic exercise leads to a post-exercise improvement in cognitive function (CF). Moreover, participants used in the published literature are not representative of the racial make-up of sport or tactical populations. Methods: A randomized crossover design was incorporated, with participants randomly consuming water or a carbohydrate sports drink within the first 3 min of a graded maximal exercise test (GMET) conducted in a laboratory. Twelve self-identified African American participants, (seven males, five females, age = 21.42 ± 2.38 years, height = 174.94 ± 12.55 cm, mass = 82.45 ± 33.09 kg) completed both testing days. Participants completed the CF tests immediately pre- and post-GMET. CF was assessed with the Stroop color and word task (SCWT) and concentration task grid (CTG). Participants completed the GMET when they reported a score of 20 on the Borg ratings of perceived exertion scale. Results: Time to complete the SCWT incongruent task (*p* < 0.001) and CTG performance (*p* < 0.001) significantly improved post-GMET in both conditions. VO_2max_ was positively correlated with pre- and post-GMET SCWT performance. Conclusions: The findings of our study suggest that an acute bout of maximal exercise significantly improves CF. Additionally, cardiorespiratory fitness is positively associated with CF in our sample of student athletes from a historically Black college and university.

## 1. Introduction

The ability to maintain high levels of cognitive functioning following fatigue from exercise at near maximal and maximal intensities is desirable for athletes. Reduced levels of cognitive functioning when fatigued after maximal physical effort could potentially impact in-game performance such as increased turnovers in basketball, unforced errors in tennis, or offside penalties in a fast-paced offense [1,2]. Tactical athletes share cognitive requirements similar to sports athletes, but often perform in situations with much higher stakes, where decreased cognitive functioning could result in mission failure, injury, or death [3]. Therefore, it is important to understand how attentional focus, an aspect of cognitive function, changes after an acute near maximal bout of exercise. Beneficial effects of improved cardiorespiratory fitness (CRF) on cognitive function have been previously observed [4,5,6] but the acute effects of a single, short-term bout of high-intensity aerobic exercise on cognitive function are not as clear [1,7,8]. As Basso and Suzuki (2017) noted in a review, there are diverse responses, from improved to detrimental, in regard to cognitive function post-exercise [7].

As concluded by several meta-analyses, a short-term, intense bout of both aerobic and anaerobic exercise can have an acute positive effect on cognitive function [1,7]. Specifically, numerous studies found positive effects on cognitive tasks that rely primarily on the pre-frontal cortex (i.e., attention and perception tasks that focus primarily on reaction time) [7,9,10,11]. The studies reporting a positive effect of acute aerobic exercise on cognitive function indicate that there is an optimal zone for cognitive function between 40 and 80% of VO_2max_, or moderate to vigorous intensity [12]. However, several studies have reported contradictory findings that acute bouts of exercise have a negative effect on aspects of cognitive functioning. For example, Thomson and colleagues reported soccer, basketball, and volleyball athletes’ decreased decision-making time but increased decision-making errors after a maximal aerobic exercise task [13]. Another study found that Reserve Officers’ Training Corps (ROTC) cadets’ cognitive function performance significantly declined during a graded exercise test while wearing their duty uniform and gear [14]. In a sport application setting, increasing reaction time at the expense of accuracy can be detrimental, and in tactical athletes, it can be fatal. 

Inconsistent findings from studies investigating the effects of acute exercise on cognitive function are likely due to different exercise protocols and cognitive tasks employed in previous research. Previous research exercise protocols varied in intensity and duration, both of which can have differing effects on the outcome of cognitive function tasks [12]. Likewise, cognitive tasks are designed to test different types of cognitive function; some familiar to a sport, such as speed to locate a soccer ball, while others are less familiar to participants, such as the Stroop color and word task [9,15]. Furthermore, previous research has used a wide variety of ages, sex, and ethnicities when analyzing the effects of exercise on cognitive function. Notably, very few studies have analyzed a racial minority population [6]. Differences in exercise intensity and duration, cognitive tasks employed, and population demographics may positively or negatively affect how exercise impacts cognitive function. More research is needed to elucidate how each of these factors individually affects cognitive function outcomes.

Another potentially confounding factor, which may have contributed to inconsistent results, is that high-intensity exercise rapidly depletes glycogen stores [16]. Several studies have reported that low glucose levels are associated with impaired aspects of cognitive functioning [17,18]. Not unexpectedly, there is substantial evidence that acute carbohydrate (CHO) intake prior to exercise improves cognitive task performance [12,19,20]. However, evidence of ingesting CHO during exercise has been mixed [19]. Although the mixed results of CHO ingestion during exercise may be due to methodological differences [19], evidence suggests that to benefit from the ergogenic effects of CHO on cognition, CHO should be consumed 1–2 h prior to exercise [19,21,22]. However, several researchers have examined the impact of a CHO mouth rinse during exercise on moderate and high-intensity exercise lasting < 1 h and reported mixed results [23]. The findings suggest that solutions that are 6–8% CHO can have positive benefits that last 20–45 min. However, the results of mouth rinsing on cognitive function are limited [23]. Recently, de Pauw and colleagues [24] examined the impact of CHO rinsing on performance in the Stroop color and word task test, with their findings suggesting that although there were no significant differences in scores, there was increased cortical activity with CHO rinsing. A limitation of the study by de Pauw and colleagues [24] was the interaction of CHO rinsing, exercise, and cognition was not examined. 

Furthermore, to our knowledge, very few studies exist that report on the effects of exercise bouts on cognitive function in young, healthy, adult Black, Indigenous, and people of color (BIPOC) populations [6]. Since sports teams and the tactical population are racially diverse, examining a minority sample may help illuminate findings specific to BIPOC populations who may have been underrepresented in previous literature [25]. The primary aim of this study was to determine the acute effects of high-intensity aerobic exercise on cognitive function in a sample of young, healthy BIPOC adults. Based on the previous literature, we hypothesized that an acute bout of high-intensity aerobic exercise would result in small improvements in cognitive function immediately post-exercise. A secondary aim was to determine whether an 8% CHO beverage consumed during exercise would affect cognitive function immediately post-exercise. Ergogenic aids to increase the physiological and cognitive performance of athletes, especially those in physically and tactically demanding sports such as soccer, tennis, biathlon, and American football, have been well researched [20,23,26,27,28].

## 2. Methods

### 2.1. Research Design

This study employed a randomized controlled crossover design where participants visited the laboratory on three occasions: (1) familiarization (~10 min), (2) testing day one (~30 min), and (3) testing day two (~30 min) (Figure 1). Testing was scheduled between 9 a.m. and 6 p.m. on a Saturday. A time for all 3 days was chosen and agreed upon by each participant. Participants were instructed to refrain from caffeine and alcohol consumption the night before the two days of testing, and were advised to get their typical amount of sleep. To account for potential diurnal variations [29], participants were scheduled ±30 min of their familiarization day. Testing days were scheduled 7 days apart. All testing was conducted in a temperature-controlled room (22.22 °C). Using randomizer.org, participants were randomly assigned the beverage they would consume each testing day. 

Since sleep loss has a substantial effect on cognition [29], participants who reported two hours more or less than their usual sleep duration were not tested that day. Sleep disturbances and prior night’s sleep were assessed to ensure these factors did not influence our results, as the prior literature has found that sleep impacts human performance [29]. The average self-reported nightly sleep during the month prior to the study was 7.2 ± 0.6 h. The number of hours of self-reported sleep the night before each of the two testing days did not significantly differ between beverage conditions (water: 7.1 ± 0.8 h vs. CHO beverage: 7.2 ± 0.7 h: *p* = 0.861). To reduce the risk of experimental error, on the familiarization day, participants were invited to spend 10 min in the lab, where they performed the cognitive tasks. None of the scores from the completion of these tests were recorded. 

### 2.2. Subjects

A convenience sample of 24 BIPOC students (16 men, 8 women) was recruited from a small historically Black college and university institution (HBCU) in a large metropolitan area in the southern United States. Potential participants were recruited from (i) large university classes (>30 students), (ii) announcements in the Exercise Science building, and (iii) through word of mouth. All individuals who completed testing were Division 1 athletes in various sports. Prior to data collection, participants completed an online screening questionnaire administered using SurveyMonkey Inc. (San Mateo, CA, USA, www.surveymonkey.com). Participant inclusion criteria included the following: (i) over the age of 18 and under the age of 27, (ii) medically cleared by a physician to complete a Bruce protocol graded maximal exercise test (GMET), (iii) no lower extremity injuries within the last 6 months, (iv) no neurological conditions, and (v) no pulmonary conditions. Participants were given a health history questionnaire to ensure that they did not have known medical conditions that could influence their ability to exercise. Participants in this study were treated in accordance with the ethical guidelines set forth by the American Psychological Association. Participants were informed of their right to withdraw from the study at any time without penalty. Participants’ anonymity and confidentiality were strictly protected, and any personal information gathered during the study was kept confidential. The study was conducted in a manner that avoided any harm or discomfort to the participants. Any risks associated with participation in the study were explained to the participants in advance. All participants read and signed the approved informed consent form prior to participation. Approval for this study was granted by Tennessee State University Institutional Review Board (approval #HS2011-2814). Due to the mixed findings of previous studies, an a priori power analysis was not conducted for this exploratory study [30], and the researchers chose to collect data until 12 participants had completed the study. 

### 2.3. Procedures

Upon reporting to lab, participants completed an initial screening to ensure that they had followed pre-testing instructions. Participants completed a series of surveys that asked them about their previous night’s sleep, and food, beverage, and drug consumption over the last 24 h [31,32,33]. Participants were screened for eligibility and then were administered a dynamic warm-up. Participants were then asked to be to complete the SCWT and the CTG while seated. After completion of the cognitive tasks, blood glucose was measured. Participants then completed the GMET using the Bruce protocol, and continued until their Borg rating of perceived exertion (RPE) was 20. The estimated VO_2max_ score was calculated using the following formulae: men = 14.8 − (1.379 × time) + (0.451 × time^2^) − (0.012 × time^3^), females = 4.38 × time − 3.9, where “time” was the total graded maximal exercise test time to completion [34]. Participants consumed their selected beverage for that testing session (either water or a CHO drink) during the first stage of the maximal exercise test. Since the Pomportes and colleagues [23] review was not published prior to data collection for this study, in order to examine the impact of the 8% CHO beverage rinsing, participants were asked to slowly sip the beverage over the course of the 3 min warmup and to ensure that every sip “sat” in their mouth prior to swallowing. After completion of the maximal exercise test, participants completed the SCWT and CTG again. Blood glucose was measured and then their body mass was recorded. No compensation was offered for participation in this study.

### 2.4. Measures

*Stroop Color and Word Test*: Executive functioning and inhibition were measured using the Stroop color and word task (SCWT). Participants were asked to identify 56 different words and colors. The colors and words were split into congruent (e.g., the word RED appeared in the color red) and incongruent (e.g., the word RED appeared in the color green). A laminated standard sized (8.5 × 11 in) piece of paper with Times New Roman 12-point font showing 56 different words and 6 different colors (yellow, blue, orange, green, red, and purple) was handed to the participant, and the words were read out loud. Two researchers were responsible for counting the number of incorrect responses and the time was recorded by 2 separate research assistants. The average time was calculated and recorded, and there were no discrepancies between the two researchers for the number of incorrect responses. The first SCWT administered was the congruent test and the second test was the incongruent test [35]. A total of 5 different congruent and incongruent grids were created. Each pre- and post-test had its own unique grid. The time and number of incorrect responses were measured by two researchers for both the congruent and in-congruent tests. The SCWT was chosen because previous studies have suggested that the presence of glucose in the oral cavity moderates the impact of self-control resources [36,37], which is measured by the SCWT. Additionally, the SCWT has been used in previous studies examining the impact of oral glucose on cognition [24,38].

*Concentration Task Grid*: The concentration task grid (CTG) was used to assess concentration and visual scanning speed [39]. The concentration grid used in this study consisted of a grid of 100 squares arranged in a 10 × 10 square. Two-digit numbers from 00 to 99 were placed randomly in the center of each square. The participants were asked to mark off as many consecutive numbers (always starting from 00) as possible within 1 min [40]. A total of 5 different grids were created and each pre- and post-test had its own unique grid.

*Blood glucose*: Blood glucose was measured using a glucose meter (Bayer Contour 7151B). 

*Body mass*: Body mass was measured using a digital scale (Tanita Total Body Composition Analyzer model TBF-410GS, Tanita Corporation, Tokyo, Japan). 

*Height*: Participants’ height was measured with a standard stadiometer and measured to the nearest 0.1 cm. 

### 2.5. Statistical Analyses

#### 2.5.1. Pre-Processing of Data

Data were entered into Microsoft Excel 15.30 (Microsoft Corporation, Redmond, WA, USA) by 2 independent researchers, and a third researcher verified that all data matched. All data were analyzed and figures made in R (R Core Team, Vienna, Austria). A combination of histograms and Shapiro–Wilk tests for normality were used to test for the distribution of data. 

#### 2.5.2. Primary Analyses

Data were not normally distributed. Thus, to assess the main effects of the GMET on blood glucose and cognitive function, a Wilcoxon signed-ranks test was used. We evaluated pre- and post-GMET differences in blood glucose and cognitive task performance (Table 1). The Wilcoxon effect size “r” is reported with interpretation, such that 0.10–<0.3 is a small effect, 0.30–<0.5 is a moderate effect, and ≥0.5 is a large effect. Additionally, the Wilcoxon signed-rank test was used to analyze within-beverage differences from pre- and post-GMET glucose and cognitive performance. To compare beverages, the absolute change from pre- to post-GMET was calculated per person for each cognitive function test and beverage condition. The mean pre- to post-GMET difference in the water condition was compared to that of the CHO beverage condition using a Mann–Whitney U test. To understand the association between VO_2max_, blood glucose, and cognitive task performance, a Spearman Rho correlation analysis was used. α = 0.05.

## 3. Results

Twelve participants, (seven men, five women, age = 21.42 ± 2.38 years, height = 174.94 ± 12.55 cm, mass = 82.45 ± 33.09 kg) completed both testing days of this study. Participants’ GMET time was 10.67 ± 2.23 min with a calculated VO_2max_ of 39.18 ± 8.97 mL/kg/min. 

### 3.1. Differences in Blood Glucose and Cognitive Task Performance Pre- and Post-Maximal Exercise

The Wilcoxon signed-rank test revealed a small to moderate significant improvement in performance on the incongruent task of the SCWT and the concentration task grid. Additionally, there was a large increase in blood glucose levels pre- and post-maximal exercise (Table 1). Interestingly, there was a significant small decline in performance on the congruent task of the SCWT. 

### 3.2. Difference between Beverages

There was no significant difference between beverages for maximal exercise time or calculated VO_2max_. Blood glucose was significantly greater post-GMET in both water and CHO beverage conditions (Table 2). However, the CHO beverage condition resulted in a significantly greater change in blood glucose compared to water. Time to complete the congruent task of the SCWT was significantly greater post-GMET in both conditions, and the CHO beverage resulted in a significantly greater increase compared to water. The incongruent task time of the SCWT significantly decreased post-GMET in both conditions. The water condition experienced a significantly greater decrease, and thus improvement, in incongruent task time, and this difference exhibits a large effect size. Incorrect answers on the incongruent task of the SCWT were significantly higher post-GMET within the water condition. The numbers found on the CTG were significantly greater post-GMET in both conditions, with the water condition displaying a significantly greater improvement with a small effect. 

### 3.3. Association between VO_2max_, Cognitive Task Performance, and Blood Glucose

The results from the Spearman Rho correlation analysis suggest that those with higher VO_2max_ had lower completion times pre-test on the SCWT incongruent task (rho = −0.471, *p* = 0.020; Figure 2), fewer incorrect responses on the pre-test incongruent task of the SCWT (r = −0.476, *p* = 0.019), lower completion times post-test on the SCWT incongruent task (rho= −0.561, *p* = 0.004; Figure 2), and higher glucose post-maximal exercise test (rho = 0.550, *p* = 0.005). Higher blood glucose levels post-maximal exercise test were associated with lower completion times on the post-test incongruent task of the SCWT (rho = −0.480, *p* = 0.018) and fewer incorrect responses on the post-test incongruent task of the SCWT (rho = −0.432, *p* = 0.035). There was no significant association (*p* > 0.05) between pre-test blood glucose level and any of the cognitive tasks.

## 4. Discussion

The purpose of this study was to examine whether a maximal aerobic exercise test would lead to acute improvements in cognitive functioning in a sample of young, healthy BIPOC adults. The main findings of this study were that some, but not all, aspects of cognitive function were significantly better post-GMET compared to pre-exercise, with moderate effect sizes. Moreover, CRF appears to be related to cognitive function, as both pre-GMET and post-GMET performance on the incongruent portion of the SCWT were significantly and moderately correlated with VO_2max_. Interestingly, no significant correlation was found between VO_2max_ and the CTG, suggesting that some cognitive function tasks may be more strongly related to CRF than others. However, we found no ergogenic effect on cognitive task performance because of CHO rinsing compared to water intake. Similarly, there was no effect of CHO rinsing for maximal exercise test time and estimated VO_2max_. The analysis of blood glucose levels revealed significantly higher blood glucose at the post-GMET timepoint, and correlations suggest that higher blood glucose was associated with better performance on the SCWT incongruent task. Likewise, higher VO_2max_ was associated with higher blood glucose post-test. To our knowledge, this is the first study to use a BIPOC population to measure changes in cognitive task performance after a graded maximal exercise test.

Several of the findings of the present study related to cognitive function are congruent with some of the existing literature. McMorris and Graydon reported that performance in detecting the presence or absence of a soccer ball was significantly better during maximal exercise, suggesting that exercising at 70% and 100% of maximum power output improved the speed of visual search [9]. The CTG is designed to assess visual scanning speed, and the present study also found improved visual scanning speed with higher post-CTG scores. Although not by design, all participants in the present study were Division 1 student athletes. This unique sample may have biased our results. Other studies have also reported positive effects of exercise on cognitive function, but the differing exercise modes and intensities, as well as differing cognitive function tasks, can make them difficult to compare. What is consistent is that exercise appears to be beneficial for cognitive tasks that involve the prefrontal cortex [7]. The SCWT used in the present study, attention concentration task with soldiers [10], and the Digit span task used to assess cognitive function in middle-aged adults [11] are considered tasks that are prefrontal cortex-dependent [7], and all showed improvements post-exercise despite differing exercise protocols and populations. 

In contrast, Thomson and colleagues found that basketball, soccer, and volleyball athletes made decisions quicker on a speed discrimination task but at the expense of accuracy following a maximal exercise test [13]. The speed discrimination task requires fast response times, approximately 0.5 s, following the presentation and discrimination of a “fast” or “slow” stimulus. In contrast, the SCWT and CTG tests in the present study lasted between 20 and 60 s. Similarly, however, participants exhibited significantly more incorrect answers on the incongruent SCWT post-GMET in the present study. In this instance, participants may also have been sacrificing accuracy for speed. Additionally, in contrast to some of our findings, ROTC cadets performed significantly worse on an executive function cognitive test at 80% and 100% heart rate reserve during a graded maximal exercise test [14]. However, during their maximal exercise test, cadets wore their combat uniform pants and 16 kg of combat armor. The added load carriage component of this study may have contributed to the differing results from our current study. Load carriage has been shown to contribute to cognitive function decline during longer periods of exercise [41,42]. We hypothesize this may be due to a shift in attentional focus to the load rather than tasks requiring cognition. However, there is no literature yet to support this. Nevertheless, this should be considered when generalizing results to tactical populations. 

Higher estimated VO_2max_ values were correlated with lower completion times post-test on the SCWT incongruent task, lower completion times pre-test on the SCWT, and fewer incorrect responses on the incongruent task of the SCWT. In the previous literature, a higher VO_2peak_ has been associated with faster reactions times and better performance on attention, working memory, and problem-solving tasks [4,5,6,43,44]. Likewise, training protocols aimed at improving CRF have been successful in improving cognitive function from pre- to post-test in both active duty airmen and young, healthy adult volunteers [15,45]. A meta-analytic review of randomized control trials concluded that chronic aerobic exercise was associated with improvements in executive function and processing speed [46]. The participants in this study, among many other populations, reap cognitive benefits from higher CRF and physical activity levels [46,47].

Interestingly, CHO rinsing did not significantly impact performance on any cognitive tasks in our protocol or on maximal exercise test time to completion. However, a CHO beverage did result in significantly higher post-GMET blood glucose levels, when compared to water. CHO may have had no effect on exercise time or calculated VO_2max_ in this study because of the relatively short nature of the exercise protocol. Despite reaching a maximum exercise level, the average time to completion was approximately ten minutes across all trials (range: 7.83–14.15 min). CHO has been successfully used as an ergogenic aid to attenuate fatigue and hypoglycemia in sporting practices and matches lasting 1–2 h [19,22]. Therefore, the full ergogenic effects of CHO consumption may have not be realized over such a short exercise period. Another contributing factor may be the timing of when CHO was provided to participants. It is possible that consuming the CHO beverage during the first stage of the exercise protocol limited CHO absorption as compared to 20 min before the test. This is supported by no significant change in blood glucose levels from pre- to post-test. Singh and colleagues gave CHO to subjects 20 min before a graded maximal exercise test and observed higher blood glucose levels post-exercise in the CHO group versus the water group [48]. Another possible explanation for the results may be that the “rinsing” protocol used by this study was not the ideal rinsing protocol as outlined by Pomportes and colleagues [23], which suggests that, ideally, participants should hold the CHO beverage in their mouth for a minimum of 5 s prior to swallowing. Our study was conducted prior to the publication of this review [23], and in our study, we asked our subjects to ensure that the beverage “sat” in their mouth prior to swallowing.

Few studies to date have incorporated an all-Black or all-BIPOC population when analyzing exercise’s effects on cognitive function [6]. The current study’s sample population was seven male and five female BIPOC Division 1 student athletes. The National Collegiate Athletic Association (NCAA) self-reports that 18.35% of their Division 1 Football Bowl Subdivision Autonomy Five Conference athletes in 2020 were Black; the second largest racial group behind white (57.33%) [49]. However, Black athletes make up approximately 60% of NCAA basketball and football teams when these teams are analyzed separately [50]. Thus, the use of a sample population representative of the races of all NCAA athletes (18%) or a typical university student population (1–4%) is not recommended for generalizability to basketball and football teams and athletes at HBCUs [50]. Therefore, our results can be generalized to the BIPOC athlete population more than previous studies on post-exercise cognitive function. While congruent with the previous literature showing that cognitive function is acutely enhanced post-exercise, the failure of the carbohydrate to act as an ergogenic aid after an acute bout of high intensity exercise is unique. 

The current study has limitations that affect the generalizability of the results. The study recruited students regardless of race; however, due to the university being an HBCU, the sample population that finished the study was entirely BIPOC students who self-identified as being African American. Additionally, the participants who completed all three days of the study were Division 1 student athletes. The NCAA sport each athlete participated in at the time varied. Thus, the results of the current study can only be generalized to young BIPOC student athletes. The VO_2max_ of each participant was calculated, rather than measured, and calculated values may be different than each participant’s true VO_2max_. However, during the preliminary review of the data, the time to completion of the GMET was not significantly different between beverage conditions. Thus, we assume that similar effort was exerted by subjects, which was sufficient for the main aim of the present study. Despite using random assignment, 9 of the 12 participants who completed this study were selected for the “water” group as their first trial. Lastly, based on previously mentioned literature [48], we recommend that future studies should provide the CHO beverage 20 min or longer before the start of the VO_2max_ test. Additionally, if using a rinsing protocol, future studies should ask participants to hold the CHO beverage in their mouth for a minimum of 5 s prior to swallowing to better assess the effectiveness of this ergogenic aid on VO_2max_ and cognitive function. 

Increasing CRF in athletes is a priority due to enhanced recovery during intermittent exercise and athlete longevity [51,52]. Furthermore, executive function may also improve with increases in CRF. Accordingly, this supports the importance of incorporating CRF training for athletes in sports considered ‘anaerobic’ or individuals in tactical occupations where executive function is integral. Overall, group means showed increases in the CTG and incongruent task time of the SCWT after the graded maximal exercise test. Thus, several measures of cognitive function improved on average. However, several participants’ cognitive function scores did decline at the post-GMET timepoint. Thus, coaches should consider using cognitive assessments post-training to better understand the influence of fatigue on their athletes’ decision-making processes and to identify those who may display impaired cognitive function under fatigue. 

## 5. Conclusions

In conclusion, our study adds to the growing body of literature showing that improved CRF levels are associated with improved cognitive function and that an acute bout of maximal exercise improves cognitive function. The primary findings of this study are that several cognitive function tasks that rely on the prefrontal cortex are performed significantly better after a single bout of maximal exercise, and higher calculated VO_2max_ is correlated with better cognitive function pre- and post-maximal exercise. We found that carbohydrate consumption/rinsing during an acute bout of maximal exercise may not serve the original intended purpose of improving performance. Notably, our sample was comprised solely of BIPOC student athletes self-identifying as African American. Future research should attempt to replicate our findings in homogenous and heterogenous samples to better understand the effects of acute maximal exercise on cognitive function and whether demographic variables (i.e., race, age, etc.) influence these effects. 

## Figures and Tables

**Figure 1 sports-11-00088-f001:**
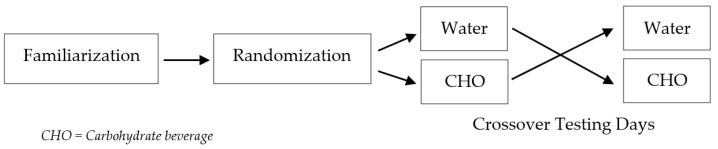
Study design.

**Figure 2 sports-11-00088-f002:**
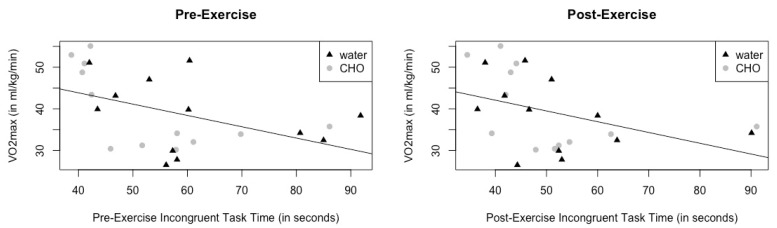
Pre- and post-exercise incongruent task time vs. calculated VO_2max._

**Table 1 sports-11-00088-t001:** Main effect of exercise on cognitive function—comparison of cognitive function pre- and post-GMET.

Variable	Pre Test	Post Test	*p*-Value	r
Glucose (mg/dL)	89.50 (22.00)	114.5 (32.50)	<0.001	0.542
CON time (s)	22.45 (2.78)	24.80 (4.50)	<0.001	0.266
INC time (s)	56.70 (17.35)	47.25 (11.43)	<0.001	0.223
INC #	2.00 (5.25)	2.00 (3.00)	0.004	0.010
CTG	17.50 (4.00)	19.00 (4.25)	<0.001	0.388

Key: Data are presented as median (interquartile range). CON = congruent task, INC = incongruent task, INC # = number of incorrect responses on incongruent SCWT, CTG = concentration task grid. r = Wilcoxon effect size.

**Table 2 sports-11-00088-t002:** Change statistics—comparison of change within and between beverage conditions from pre- to post-GMET.

Variable	Water	CHO Beverage	*p*-Value	r
Glucose (mg/dL)	18.00 (17.50) *	32.50 (45.50) *	<0.001	0.215
CON time (s)	0.45 (3.98) *	2.95 (3.42) *	<0.001	0.368
INC time (s)	−6.05 (9.18) *	−0.80 (9.23) *	0.03	0.444
INC #	0.00 (7.5) *	0.00 (2.25)	0.002	0.118
CTG	4.00 (4.50) *	2.00 (6.75) *	<0.001	0.139

Data are presented as median (interquartile range) of pre- to post-GMET difference. * indicates that the within-beverage change pre- to post-GMET is significant. Key: CON = congruent task, INC = incongruent task, INC # = number of incorrect responses on incongruent SCWT, CTG = concentration task grid. r = Wilcoxon effect size for an unpaired comparison.

## Data Availability

The data presented in this study are available on request from the corresponding author. The data are not publicly available due to the IRB limitations.

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
