# Peer review of "An Intense Bout of Acute Aerobic Exercise, but Not a Carbohydrate Supplement, Improves Cognitive Task Performance in a Sample of Black, Indigenous, and People of Color (BIPOC) Student Athletes"

_sports, 2023, doi:10.3390/sports11040088_

Round 1

Reviewer 1 Report

First of all, I would like to thank you for inviting me to evaluate this paper. I read your work thoroughly. Thanking you for your efforts, I personally think that the article was written very careless and fast. I recommend that all authors re-read and rewrite the article from beginning to end. The introduction is very complex and does not adequately defend why the study was done. The CHO intake part is not emphasized enough. The method part is very difficult to follow and the statistics part is written very sloppy. It's not in a condition that can be fixed even with a major revision. So I think the article should be rewritten from start to finish.

Author Response

We are pleased to resubmit our manuscript and believe we have fully addressed the comments provided by the editor and reviewers. Additionally, we would like to express our appreciation to each of the reviewers for providing the feedback. Below you will find the reviewers’ concerns and our answers in red.

Reviewer #1:

First of all, I would like to thank you for inviting me to evaluate this paper. I read your work thoroughly. Thanking you for your efforts, I personally think that the article was written very careless and fast. I recommend that all authors re-read and rewrite the article from beginning to end. The introduction is very complex and does not adequately defend why the study was done. The CHO intake part is not emphasized enough. The method part is very difficult to follow and the statistics part is written very sloppy. It's not in a condition that can be fixed even with a major revision. So I think the article should be rewritten from start to finish.

Response: Thank you for taking the time to read our manuscript and provide this feedback. We recognize that our methods section needed to be cleaned up and have done so based on your feedback and that of other reviewers. Substantially revisions were made to the introduction as well. However, we must respectfully disagree that the whole manuscript needs to be re-written as three other reviewers and the editor did not share the same sentiments regarding the overall writing of the manuscript.

Reviewer 2 Report

3. On the title use the complete term, not the abbreviation. Moreover, I suggest using the BIPOC. This abbreviation describes better your sample.

12. Do you mean post–training?

16. 7 males – 5 females

25. Please, explain what HBCU means.

34. Please, check the format of the literature in the text. The first reference is numbers 9 and 35.

70-80. This part of the introduction includes a topic that is not described on the title. If you examined both acute high-intensity training and the effects of CHO, you must report it on title.

90. Is the innovation of your research that you used this type of training and CHO beverages on the BIPOC sample?

112. “a priori”

112- 113. Use a reference about the priori analysis.

116. How sleep quality was assessed?

122-123. This sentence could be a part of the “procedures” of your study.

166. In sentence 122 you reported that the participants refrained from caffeine and exercise. Here, you present that the participants who consumed foods or beverages containing caffeine were not evaluated.

170. This information fits in the measurements section.

175-176. Similarly, this information fits in the measurements section.

188. I don’t understand the sentence.

199. Did you use Shapiro – wilk test for normality? If yes, did you find normality following parametric analysis?

Generally, your methodology needs extensive corrections. It was difficult for me to follow the flow of the measurements and the information of the participants. Clarify the content of each section and don’t mix the information. Also, I didn’t read clearly the cross-over design. A figure with the process would be helpful.

227 & 232. I noticed three different values of Glucose. I expected two (before – after water and glucose ingestion, respectively). Can you explain what describes each of them?

232. Delete mass from that table. Also, INC # I don’t understand if its one parameter or two.

245. The diagram is incorrect. You have put the same for both pre and post-conditions.

344. It’s the first time in the text that I read about the participants’ athletic level. Add this information to the methodology.

Author Response

We are pleased to resubmit our manuscript and believe we have fully addressed the comments provided by the editor and reviewers. Additionally, we would like to express our appreciation to each of the reviewers for providing the feedback. Below you will find the reviewers’ concerns and our answers in red.

Reviewer #2:

Line 3. On the title use the complete term, not the abbreviation. Moreover, I suggest using the BIPOC. This abbreviation describes better your sample.

Response: Thank you for your feedback. We have changed the title to be BIPOC student-athletes.

Line 12. Do you mean post–training?

Response: In this context, we are referring to any post-exercise session. We have changed it to “post-exercise” to remove confusion.

Line 16. 7 males – 5 females

Response: We have changed men and women to males and females.

Line 25. Please, explain what HBCU means.

Response: We have spelled out HBCU in the abstract and have also added on Line 16 that these student athletes self-identified as African American.

Line 34. Please, check the format of the literature in the text. The first reference is numbers 9 and 35.

Response: We appreciate the reviewer’s comment however, on our end the references are correctly labeled. We will work with the editor to ensure that the references are correctly formatted.

Lines 70-80. This part of the introduction includes a topic that is not described on the title. If you examined both acute high-intensity training and the effects of CHO, you must report it on title.

Response: Thank you for pointing this out. We originally omitted this from the title because the carbohydrate supplement had no significant effect on physical or cognitive performance. Additionally, we have left it out of the title to reduce the length. It has been added back into the title to further clarify what was done in this study.

Line 90. Is the innovation of your research that you used this type of training and CHO beverages on the BIPOC sample?

Response: We believe that our research is innovative because it includes a population that is comprised entirely of BIPOC participants. This has not been done yet when analyzing how a carbohydrate supplement and an acute bout of exercise effects cognitive function. We believe this is important given that many athletes at the college and professional level are BIPOC and yet, have not been studied.

Line 112. “a priori” 

Response: We have corrected this error.

Lines 112- 113. Use a reference about the priori analysis.

Response: Thank you for the feedback. We have added the following reference:

Kang, H. Sample Size Determination and Power Analysis Using the G*Power Software. J Educ Eval Health Prof 2021, 18, 17, doi:10.3352/jeehp.2021.18.17.

Line 116. How sleep quality was assessed?

Response: We apologize for this error. In our study sleep disturbances were assessed when asking about prior night’s sleep. We asked participants how many times they had woken up in the middle of the night and how long they were up total in the middle of the night. These numbers were used to calculate prior night’s sleep. We have now made those edits in our methodology section.

122-123. This sentence could be a part of the “procedures” of your study.

Response: Thank you for pointing this out. We have removed the “refrain from caffeine” sentence as this is covered later on and redundant. We have moved the “no compensation was offered” sentence down to the end of the procedures section.

Line 166. In sentence 122 you reported that the participants refrained from caffeine and exercise. Here, you present that the participants who consumed foods or beverages containing caffeine were not evaluated.

Response: We apologize as this may seem redundant. This was included to emphasize that participants were instructed not to consume caffeine. However, if they still showed up for testing day and self-reported that they consumed caffeine or drugs, they were not tested. We have removed this as it is somewhat redundant and may lead to confusion.

Line 170. This information fits in the measurements section.

Response: We agree that the height fits better in the measurements section. This has been added on line 206, below body mass.

Line 175-176. Similarly, this information fits in the measurements section.

Response: We agree that this information is redundant. As we have already discussed how body mass was measured in the measurements section, this has been removed.

Line 188. I don’t understand the sentence.

Response: We apologize for the confusion. We have updated the sentence to make it more understandable. Participants’ blood glucose was measured and then their body mass was taken.

Line 199: Did you use Shapiro – wilk test for normality? If yes, did you find normality following parametric analysis?

Response: We used the Shapiro-Wilk to test for normality. We also visualized these data histograms. The data was not normal, and we tried various normalization techniques but were unable to normalize the data as data at all time points would have needed the same normalization technique. However, due to there not being a non-parametric alternative to a 2X2 repeated measures ANOVA, we conducted a parametric analysis. However, all post-hoc analyses were non-parametric.

Generally, your methodology needs extensive corrections. It was difficult for me to follow the flow of the measurements and the information of the participants. Clarify the content of each section and don’t mix the information. Also, I didn’t read clearly the cross-over design. A figure with the process would be helpful

Response: We apologize for the confusion in the methods section. This has been adjusted to have a better flow of how the measurements were conducted. Additionally, we have clarified (Lines 110-119) about how the crossover occurred. We have also added a new figure (Figure 1) to the methodology.

Lines 227 & 232. I noticed three different values of Glucose. I expected two (before – after water and glucose ingestion, respectively). Can you explain what describes each of them?

Response: Thank you for pointing this out. Because it was a 2x2 RM ANOVA (test timepoint * beverage), the data presented in table 1 is for the main effects of test time point. Not considering beverage consumed, that was the main effect of test time point on blood glucose and cognitive function. The main effect of test time point was included in a separate table as these were our only significant results. In table 2, we have included the main effect of beverage as well as the interaction effects, both of which are non-significant. To further clarify this, the titles of the tables have been adjusted.  

Line 232. Delete mass from that table. Also, INC # I don’t understand if its one parameter or two.

Response: Mass has been deleted from the table. INC # is the number of words they said incorrect on the incongruent Stroop’s Color and Word Task. We have further clarified in the key to each table what the abbreviations mean. Additionally, we have used the same abbreviations in both tables to improve them.

  1. The diagram is incorrect. You have put the same for both pre and post-conditions.

Response: We apologize for this error. We have input the correct post-exercise figure.

  1. It’s the first time in the text that I read about the participants’ athletic level. Add this information to the methodology.

Response: Thank you and we apologize for omitting this from the participant’s section of the methodology. It has been added on line 136.

Reviewer 3 Report

El título de la obra debe ser más breve. Mejorar el rendimiento de las tareas cognitivas a través del intento de ejercicio aeróbico

El resumen debe incluir el contexto donde se realiza la investigación, así como la edad de los participantes.

El tema de estudio es de interés.

El procedimiento está bien. los resultados estan bien

Author Response

We are pleased to resubmit our manuscript and believe we have fully addressed the comments provided by the editor and reviewers. Additionally, we would like to express our appreciation to each of the reviewers for providing the feedback. Below you will find the reviewers’ concerns and our answers in red.

Reviewer #3:

We thank reviewer 3 for their comments and for taking their time to review our manuscript. Our responses are based on the translation of their feedback using Google Translate. We apologize if the translation is not fully accurate.

The title of the work should be shorter. Improve the performance of cognitive tasks through the attempt of aerobic exercise

Response: We agree that the title is long. We have removed the “randomized-controlled crossover design” from the title. However, based on comments from other reviewers, have added in that we also tested a carbohydrate supplement within this study.

The abstract must include the context where the research is carried out, as well as the age of the participants.

Response: In the abstract, we have added on line 16 that the research was conducted within a laboratory. Additionally, we have added participants age, height, and mass to the abstract, on line 17, to give a better idea to readers of the demographics and anthropometrics of our sample.

The subject of study is of interest.

Response: Thank you!

The procedure is fine. The results are good

Response: Thank you! We have adjusted the methods slightly based on the feedback from other reviewers. We hope that it makes the methods section more understandable.

Reviewer 4 Report

I would like to congratulate the authors and thank them for their effort and motivation involved in this research study. The presentation of the research is well documented, with a scientific basis and respects the latest standards regarding the highest level scientific publications. The methodology was chosen correctly. The conclusions support and result from the research and open new directions for future research. The submitted work is interesting and essentially exhausts the subject under discussion. However, I do have a few minor suggestions to add to the manuscript to improve it:

1) The article states that the study was conducted in accordance with the Declaration of Helsinki, what is extremely important. However, there is no information whether the participants were treated ethically according to the American Psychological Association code of ethics? Please complete this information in the manuscript.

2) All references should be brought into line with the MDPI Instruction for Authors, which is included on the journal’s website (in particular, bring the cited items into line with MDPI and ACS Style and add DOI numbers).

3) The article needs to be corrected for typos and unclear wording. Furthermore, the article gives the impression of being unfinished; I would suggest reading it several times and fine-tuning the technical errors.

4) Line 111: what does it mean: XXXXXXX ? Please explain it.

5) No clear conclusions were extracted from the manuscript. It is suggested that the authors separate a section for the conclusion.

6) In my opinion, the title is too long and suggests an immediately obtained result. I suggest shortening the title.

Author Response

We are pleased to resubmit our manuscript and believe we have fully addressed the comments provided by the editor and reviewers. Additionally, we would like to express our appreciation to each of the reviewers for providing the feedback. Below you will find the reviewers’ concerns and our answers in red.

Reviewer #4:

I would like to congratulate the authors and thank them for their effort and motivation involved in this research study. The presentation of the research is well documented, with a scientific basis and respects the latest standards regarding the highest level scientific publications. The methodology was chosen correctly. The conclusions support and result from the research and open new directions for future research. The submitted work is interesting and essentially exhausts the subject under discussion. However, I do have a few minor suggestions to add to the manuscript to improve it:

Response: We would like to thank you for taking your time to review our manuscript and for the feedback you have provided.

1) The article states that the study was conducted in accordance with the Declaration of Helsinki, what is extremely important. However, there is no information whether the participants were treated ethically according to the American Psychological Association code of ethics? Please complete this information in the manuscript.

Response:

We apologize for omitting this important information from the study. We have added the following text:

Participants in this study were treated in accordance with the ethical guidelines set forth by the American Psychological Association. Participants were informed of their right to withdraw from the study at any time without penalty. Participants' anonymity and confidentiality was strictly protected, and any personal information gathered during the study will be kept confidential. The study was conducted in a manner that avoids any harm or discomfort to the participants. Any risks associated with participation in the study were explained to the participants in advance.

2) All references should be brought into line with the MDPI Instruction for Authors, which is included on the journal’s website (in particular, bring the cited items into line with MDPI and ACS Style and add DOI numbers).

Response: We apologize for not following the MDPI reference style initially. These have been formatted correctly in the revision.

3) The article needs to be corrected for typos and unclear wording. Furthermore, the article gives the impression of being unfinished; I would suggest reading it several times and fine-tuning the technical errors.

Response: Thank you for your feedback. We have taken time to read the manuscript and correct typographical and grammatical errors. We also have taken out portions of our methods section that are redundant in order to streamline the procedures. Likewise, we have clarified portions of our methods section to be more understandable.

4) Line 111: what does it mean: XXXXXXX ? Please explain it.

Response: We have included X’s, rather than the name of the institution that approved the IRB, to retain anonymity in review and retain your blind review. The name of the institution will be added upon acceptance.

5) No clear conclusions were extracted from the manuscript. It is suggested that the authors separate a section for the conclusion.

Response: We thank you for your comment. We have now added a section labeled “Conclusion” which summarizes the findings of our study and provides guidance for future research.

6) In my opinion, the title is too long and suggests an immediately obtained result. I suggest shortening the title.

Response: We agree that the title is too long. We have removed the “randomized-controlled crossover design” from the title. However, based on comments from other reviewers, we have added in that we also tested a carbohydrate supplement within this study. However, our results do support an immediately obtained results as cognitive function was improved immediately after a single bout of maximal exercise.

Round 2

Reviewer 1 Report

Thanks for authors adressing the suggestion of reviewers, now your manuscripts seems better. Congrats.

Author Response

Thanks for authors adressing the suggestion of reviewers, now your manuscripts seems better. Congrats.

We would like to thank the reviewer for their thoughtful review.

Reviewer 2 Report

4. Probably, I did not describe my previous comment well. It is important to use the complete term BIPOC.

224. It is not appropriate to use that kind of analysis. Check this test for non-parametric analysis: Mann-Whitney test (independent samples) or Wilcoxon test (dependent samples) for two groups. Friedman test for two and more groups and dependent samples, Kruskal-Wallis test for independent samples and two or more groups.

Author Response

We are pleased to resubmit our manuscript and believe we have fully addressed the comments provided by the editor and reviewers. Additionally, we would like to express our appreciation to each of the reviewers for providing the feedback. Below you will find the reviewers’ concerns and our answers in red.

Reviewer #2:

  1. Probably, I did not describe my previous comment well. It is important to use the complete term BIPOC.

Response: Thank you for clarifying. We have included the complete term BIPOC in the title now:

An intense bout of acute aerobic exercise, but not a carbohydrate supplement, improves cognitive task performance in a sample of Black, Indigenous, and People of Color (BIPOC) student-athletes

  1. It is not appropriate to use that kind of analysis. Check this test for non-parametric analysis: Mann-Whitney test (independent samples) or Wilcoxon test (dependent samples) for two groups. Friedman test for two and more groups and dependent samples, Kruskal-Wallis test for independent samples and two or more groups.

Response: Thank you for providing your expertise. The analysis has been re-run with all non-parametric tests. A Wilcoxon signed ranks tests was used to assess main effects of the maximal exercise on cognitive function performance. Additionally, the Wilcoxon signed ranks test was used to assess effects of exercise within beverage condition. To compare beverages, absolute change from pre- to post-exercise was calculated for each variable within each beverage. These were then compared using a Mann-Whitney U Test. Additionally, a Spearman’s Rho correlation was used in place of a Pearson’s Correlation. These change in analyses have resulted in several updates to our results. These changes have been reflected throughout the manuscript.

We thank reviewers 1 and 2 for the additional feedback. We believe it has been very helpful in improving the quality of our manuscript.
